# Robustness Certification of Visual Perception Models via Camera Motion Smoothing

**Hanjiang Hu**[1] **Zuxin Liu**[1] **Linyi Li**[2] **Jiacheng Zhu**[1] **Ding Zhao**[1]
[1]Carnegie Mellon University    [2]University of Illinois at Urbana-Champaign
{hanjianghu,dingzhao}@cmu.edu, {zuxinl,jzhu4}@andrew.cmu.edu, linyi2@illinois.edu

**Abstract:** A vast literature shows that the learning-based visual perception model is sensitive to adversarial noises but few works consider the robustness of robotic perception models under widely-existing camera motion perturbations. To this end, we study the robustness of the visual perception model under camera motion perturbations to investigate the influence of camera motion on robotic perception. Specifically, we propose a motion smoothing technique for arbitrary image classification models, whose robustness under camera motion perturbations could be certified. The proposed robustness certification framework based on camera motion smoothing provides effective and scalable robustness guarantees for visual perception modules so that they are applicable to wide robotic applications. As far as we are aware, this is the first work to provide the robustness certification for the deep perception module against camera motions, which improves the trustworthiness of robotic perception. A realistic indoor robotic dataset with the dense point cloud map for the entire room, *MetaRoom*, is introduced for the challenging certifiable robust perception task. We conduct extensive experiments to validate the certification approach via motion smoothing against camera motion perturbations. Our framework guarantees the certified accuracy of 81.7% against camera translation perturbation along depth direction within -0.1m ∼ 0.1m. We also validate the effectiveness of our method on real-world robot by conducting hardware experiment on robotic arm with an eye-in-hand camera. The code is available on https://github.com/HanjiangHu/camera-motion-smoothing.

**Keywords:** Certifiable Robustness, Camera Motion Perturbation, Perception

## 1 Introduction

Visual perception has achieved great success in recent years by leveraging the powerful representation capability of neural networks and large-scale datasets [1, 2, 3]. Deep learning models have been dominating a wide range of computer vision tasks, such as image classification [4, 5], object detection [6, 7, 8] and segmentation [9, 10, 11]. However, applying deep perception models to real-world robotic applications is still challenging. Since visual perception is the core upstream module of an autonomous robot system, its failure can cause the robot to sense the environments incorrectly, which may result in catastrophic consequences. Therefore, developing a trustworthy perception system that can guarantee functionality in diverse real-world scenarios is necessary [12, 13, 14].

We study the robustness of a visual perception system against sensing noise due to camera motion perturbation that commonly exists in robotic applications [15, 16, 14], which is important while challenging for trustworthy robotic applications. The difficulty arises from two perspectives: internal model vulnerability and external sensing uncertainty. On the one hand, rich literature suggests that deep visual models are vulnerable to adversarial perturbations that would be stealthy to human eyes: small perturbations added to the input of neural networks can significantly corrupt the perception performance [17, 18, 19, 20]. On the other hand, several recent works indicate that the perception performance is also sensitive to data acquisition, such as sensor placement and sensing perspective [21, 22, 23, 24]. Both internal vulnerability and external sensing uncertainty make it challenging to guarantee the robustness of a visual system in real-world robotic applications.

6th Conference on Robot Learning (CoRL 2022), Auckland, New Zealand.

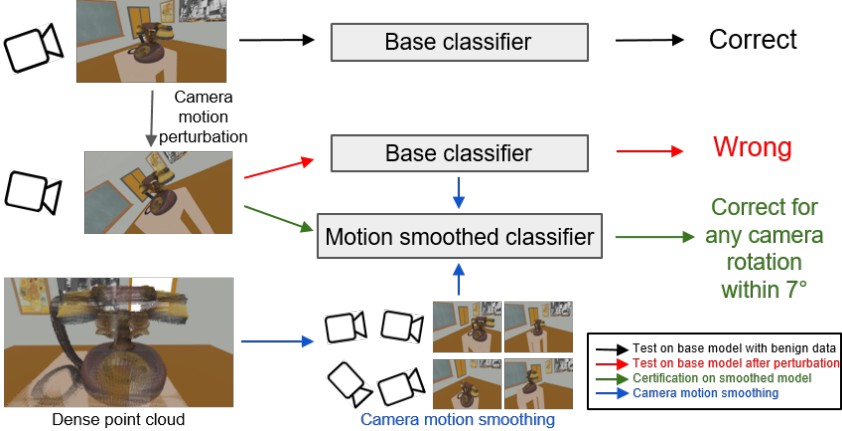

Figure 1: Certification framework via camera motion smoothing

Prior works propose several techniques to improve the visual perception system robustness [25, 26, 27], though most of them are demonstrated to be effective empirically and no theoretical robustness guarantees are given. A recent line of work aims to provide provable robustness certification or verification such that the perception system is guaranteed to function properly under any bounded adversarial perturbations [28, 29, 30] or semantic image transformations and deformations [31, 32, 33], which improves the trustworthiness of the perception model. However, most of them are studied under static 2D image datasets without viewpoint changes, while the robotic visual perception systems process 2D images projected from the 3D physical world, which may not be robust under different perspectives of the moving camera and make it challenging to certify the robustness for real-world robotic applications.

To tackle the challenge, we first study the robustness of the robotic perception model against the camera motion perturbation. Next, we propose the *first* framework with a certified robustness guarantee for robotic perception models against arbitrary bounded camera motion perturbations based on a novel motion smoothing strategy as shown in Fig. 1. Extensive experiments have been conducted on a realistic indoor robotic dataset *MetaRoom* with the dense point cloud map as an given oracle for image projection, which is collected from the Webots simulation environment [34] to show the effectiveness of the proposed robustness certification method against camera motion perturbations. To the best of our knowledge, this is the first work to study the certifiable robustness of image-based robotic perception under camera ego-motion based on motion smoothing. The contributions are summarized as follows.

- We demonstrate that the camera motion perturbation can significantly influence perception performance and introduce motion smoothing to improve robustness over camera motion perturbation.
- We propose a smoothing algorithm for any black-box image classification model such that its robustness against camera motion perturbations can be certified by our certification framework.
- We conduct extensive experiments on the realistic indoor robotic simulation to validate the effectiveness of the motion smoothing certification, achieving over 80% certified accuracy against any perturbations within radii of 0.1m or 7° for camera translation or rotation along the depth axis. Further experiments on real-world robot arm validate the effectiveness of camera motion smoothing for perception models.

## 2 Related Work

**Robust Robotic Perception.** The robustness of robotic perception has been studied from different viewpoints. A rich literature in the robust machine learning community shows that deep learning-based robotic perception models are vulnerable and can be easily fooled by adversarial samples [17, 18, 19, 20]. Another perspective for robotic perception is the external sensing uncertainty, which is caused by sensor placement [21, 23], camera distortions [24], geometric outliers [14], long-term robustness [35, 36], sim-to-real adaptation [37, 10, 38], etc. However, the robustness of deep perception models given the perturbed projected images from 3D physical world with a moving camera sensor is relatively understudied.

**Certifiable Robustness under Perturbations.** In recent years, a significant number of approaches has been proposed to provide robustness certification for deep neural networks [39, 40]. In contrast to empirical robustness approaches [41, 25, 26, 27], i.e., which train robust models against

adversarial perturbations, the robustness certification approaches aim to guarantee the accuracy of the perception model as long as the perturbation magnitude is bounded by some threshold. Such robustness certification approaches have been proposed against both $\ell_p$-bounded pixel-wise perturbations [28, 29, 30, 42, 43, 44] and semantic transformation or deformations [45, 31, 32, 33, 46], providing either deterministic guarantees based on function relaxations [46, 47, 48, 32] or high-confident probabilistic guarantees based on random smoothing [49, 31, 33, 50, 45]. However, they consider either 2D or 3D transformations. To the best of our knowledge, no prior work studies certifiable robustness associated with the movement of sensors and 3D-2D projected images, despite the fact that it is commonly seen in trustworthy robotic applications. Therefore, we aim to bridge the camera motion perturbation with deep learning robustness certification for perception systems.

## 3   Methodology

In this section, we introduce the robustness certification framework against camera motion perturbations through the motion-smoothed perception model. We first define the image projection in terms of camera motion. Then, we clarify the certification goal and define the camera motion smoothed classifier using image projection. Finally, we present the robustness certification for each decomposed translation and rotation translation.

### 3.1   Image Projection with Camera Motion

We first define the positive projection in Def. 1 based on the camera imaging concept in geometric computer vision [51]. The follow-up Def. 2 defines the relative projective transformation $\phi(x, \alpha)$ parameterized by relative camera motion $\alpha$ with respect to the image $x$ at motion origin.

**Definition 1** (Position projective function). *For any 3D point $P = (X, Y, Z) \in \mathbb{P} \subset \mathbb{R}^3$ under the camera coordinate frame with the camera intrinsic matrix $K$, based on the camera motion $\alpha = (\boldsymbol{\theta}, t) \in \mathcal{Z} \subset \mathbb{R}^6$ with rotation matrix $R = \exp(\boldsymbol{\theta}^\wedge) \in SO(3)$ and translation vector $t \in \mathbb{R}^3$, we define the position projective function $\rho : \mathbb{P} \times \mathcal{Z} \to \mathbb{R}^2$ and the depth function $D : \mathbb{P} \times \mathcal{Z} \to \mathbb{R}$ for point $P$ as*

$$[\rho(P, \alpha), 1]^\top = \frac{1}{D(P, \alpha)} K R^{-1}(P - t), \quad D(P, \alpha) = [0, 0, 1] R^{-1}(P - t) \qquad (1)$$

**Definition 2** (Channel-wise projective transformation). *Given the position projection function $\rho : \mathbb{P} \times \mathcal{Z} \to \mathbb{R}^2$ and the depth function $D : \mathbb{P} \times \mathcal{Z} \to \mathbb{R}$ over dense 3D point cloud $\mathbb{P}$, define the 3D-2D global channel-wise projective transformation from C-channel colored point cloud $\mathbb{V} = (\mathbb{R}^C, \mathbb{P}) \subset \mathbb{R}^{C+3}$ to $H \times W$ image gird $\mathcal{X} \subset \mathbb{R}^{C \times H \times W}$ as $O : \mathbb{V} \times \mathcal{Z} \to \mathcal{X}$ parameterized by camera motion $\alpha \in \mathcal{Z}$ using Floor function $\lfloor \cdot \rfloor$,*

$$x_{c,r,s} = O(V, \alpha)_{c,r,s} = V_{c, P_\alpha^*}, where \; P_\alpha^* = \underset{\{P \in \mathbb{P} | \lfloor \rho(P, \alpha) \rfloor = (r, s)\}}{\operatorname{argmin}} D(P, \alpha) \qquad (2)$$

*Specifically, if $x = O(V, 0)$, we define the relative projective transformation $\phi : \mathcal{X} \times \mathcal{Z} \to \mathcal{X}$ as $\phi(x, \alpha) = O(V, \alpha)$.*

### 3.2   Certification Goal and Motion Smoothed Classifier

**Certification Goal.** We consider the classification task as the most fundamental perception task. For any deep learning-based classification model $h$, given the projected image $x$ at the origin of camera motion in the motion space $\mathcal{Z}$, the certification goal is to find a set within a radius $\mathcal{Z}_{\text{radius}} \subseteq \mathcal{Z}$ such that, for any relative projective transformation $\phi \in \mathcal{Z}_{\text{radius}}$, with high confidence we have

$$h(x) = h(\phi(x, \alpha)), \forall \alpha \in \mathcal{Z}_{\text{radius}}. \qquad (3)$$

Based on the relative projective transformation $\phi$ over camera motion space $\mathcal{Z}$ defined above, we define the camera motion smoothed classification model as follows.

**Definition 3** (Camera motion $\varepsilon$-smoothed classifier). *Let $\phi : \mathcal{X} \times \mathcal{Z} \to \mathcal{X}$ be a relative projective transformation given the projected image $x$ at the origin of camera motion in the motion space $\mathcal{Z}$, and let $\varepsilon \sim \mathcal{P}_\varepsilon$ be a random camera motion taking values in $\mathcal{Z}$. Let $h : \mathcal{X} \to \mathcal{Y}$ be a base classifier $h(x) = \operatorname{argmax}_{y \in \mathcal{Y}} p(y \mid x)$, the expectation of projected image predictions $\phi(x, \varepsilon)$ over camera motion distribution $\mathcal{P}_\varepsilon$ is $q(y \mid x; \varepsilon) := \mathbb{E}_{\varepsilon \sim \mathcal{P}_\varepsilon} p(y \mid \phi(x, \varepsilon))$. We define the $\varepsilon$-smoothed classifier $g : \mathcal{X} \to \mathcal{Y}$ as*

$$g(x; \varepsilon) := \underset{y \in \mathcal{Y}}{\operatorname{argmax}} \, q(y \mid x; \varepsilon) = \underset{y \in \mathcal{Y}}{\operatorname{argmax}} \, \mathbb{E}_{\varepsilon \sim \mathcal{P}_\varepsilon} p(y \mid \phi(x, \varepsilon)). \qquad (4)$$

### 3.3 Certifying Motion-parameterized Projection with Smoothed Classifier

In order to achieve the certification goal 3 with smoothed classifier, prior works [31, 50, 52] show that smoothed model can be certified if the image transformation is resolvable. To this end, we first show that the relative projection is generally compatible with the global projection, which indicates that image projection can be regarded as resolvable transformation.

**Lemma 1** (Compatible Relative Projection with Global Projection). *With a global projective transformation $O : \mathbb{V} \times \mathcal{Z} \rightarrow \mathcal{X}$ from 3D point cloud and a relative projective transformation $\phi : \mathcal{X} \times \mathcal{Z} \rightarrow \mathcal{X}$ given some original camera motions, for any $\alpha_1 \in \mathcal{Z}$ there exists an injective, continuously differentiable and non-vanishing-Jacobian function $\gamma_{\alpha_1} : \mathcal{Z} \rightarrow \mathcal{Z}$ such that*

$$\phi(O(V, \alpha_1), \alpha_2) = O(V, \gamma_{\alpha_1}(\alpha_2)), V \in \mathbb{V}, \alpha_2 \in \mathcal{Z}. \tag{5}$$

The high-level idea for the proof is that $SO(3)$ has the rules of multiplication and inverse operation which can be transferred to the $\gamma_{\alpha_1}$ function (referred as $\gamma$ for convenience) in (5), and min-pooling defined in the projective transformation in (2) does not break these rules as well. The full proof can be found in the supplementary materials. Specifically, if the camera motion is with translation and fixed-axis rotation, we have the following certification theorem.

**Theorem 1** (Robustness certification under camera motion with fixed-axis rotation). *Let $\alpha \in \mathcal{Z} \subset \mathbb{R}^6$ be the parameters of projective transformation $\phi$ with translation $(t_x, t_y, t_z)^T \in \mathbb{R}^3$ and fixed-axis rotation $(\theta n_1, \theta n_2, \theta n_3)^T \in \mathbb{R}^3, \sum_{i=1}^3 n_i^2 = 1$, suppose the composed camera motion $\varepsilon_1 \in \mathcal{Z}$ satisfies $\phi(x, \varepsilon_1) = \phi(\phi(x, \varepsilon_0), \alpha)$ given some $\alpha \in \mathcal{Z}$ and zero-mean Gaussian motion $\varepsilon_0$ with variance $\sigma_x^2, \sigma_y^2, \sigma_z^2, \sigma_\theta^2$ for $t_x, t_y, t_z, \theta$ respectively, let $p_A, p_B \in [0, 1]$ be bounds of the top-2 class probabilities for the motion smoothed model, i.e.,*

$$q(y_A \mid x, \varepsilon_0) \geq p_A > p_B \geq \max_{y \neq y_A} q(y \mid x, \varepsilon_0). \tag{6}$$

*Then, it holds that $g(\phi(x, \alpha); \varepsilon_0) = g(x; \varepsilon_0)$ if $\alpha = (t_x, t_y, t_z, \theta n_1, \theta n_2, \theta n_3)^T$ satisfies*

$$\sqrt{\left(\frac{\theta}{\sigma_\theta}\right)^2 + \left(\frac{t_x}{\sigma_x}\right)^2 + \left(\frac{t_y}{\sigma_y}\right)^2 + \left(\frac{t_z}{\sigma_z}\right)^2} < \frac{1}{2}\left(\Phi^{-1}(p_A) - \Phi^{-1}(p_B)\right). \tag{7}$$

The proof sketch is that based on Lemma 1, the relative projection is resovable and compatible with the global projection, and specifically, $\gamma$ function in (5) will be additive $\gamma_{\alpha_1}(\alpha_2) = \alpha_1 + \alpha_2$ with fixed-axis rotation. The full proof can be found in the supplementary materials following previous certification work [28, 31]. We remark that for the general rotation without a fixed axis, although $\varepsilon_1 = \alpha + \varepsilon_0$ does not hold, the $\gamma$ function can also be found based on multiplication in $SO(3)$ since Lemma 1 holds in general cases.

Since all the camera motions can be regarded as the composition of each 1-axis translation or rotation, we make the following corollary to show the certification for any 1-axis translation or rotation.

**Corollary 1** (Certification of 1-axis motion perturbation). *For any 1-axis camera motion perturbation $\alpha$ with non-zero entry $\alpha_i$ satisfying $\alpha_i < \frac{\sigma_i}{2}\left(\Phi^{-1}(p_A) - \Phi^{-1}(p_B)\right)$ under motion smoothing $\varepsilon_0 \sim \mathcal{N}(0, \sigma_i^2)$, it holds that $g(\phi(x, \alpha); \varepsilon_0) = g(x; \varepsilon_0)$ for the motion smoothed classifier $g$.*

**Remark 1.** *Cor. 1 directly follows Thm. 1 by taking only one non-zero entry in $\alpha \in \mathcal{Z} \subset \mathbb{R}^6$, where each entry of in $\alpha = (t_x, t_y, t_z, \theta n_1, \theta n_2, \theta n_3)^T$ means $T_x$: translation along depth-orthogonal horizontal axis, $T_y$: translation along depth-orthogonal vertical axis, $T_z$: translation along depth axis, $R_x$: rotation around depth-orthogonal pitch axis, $R_y$: rotation around depth-orthogonal yaw axis and $R_z$: rotation around depth roll axis, as shown in Fig. 3.*

Therefore, we conduct comprehensive experiments for each 1-axis camera motion based on the Cor. 1 and Remark 1 for robustness certification.

## 4 Experiments

In this section, we aim to address two questions: 1) How does the perturbation of camera motion influence the perception performance empirically? 2) How can we certify the accuracy using the motion-smoothed model under the camera perturbation within some radius? To answer these questions, we first set up a simulated indoor environment *MetaRoom* with dense point cloud maps and conduct extensive experiments based on it. Besides, we also conduct real-world experiments on a robotic arm with an eye-in-hand camera to investigate motion smoothing in robotic applications. Supplementary materials contain more experiment details.

### 4.1 Experimental Setup

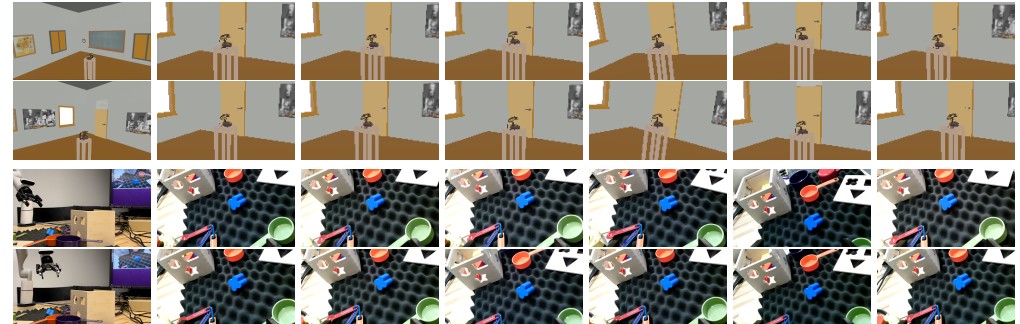

Figure 2: *MetaRoom* images under 6-axis perturbations (first two rows) and a real-world robot with an eye-in-hand camera (last two rows). Left to right: global scenarios, perturbations over $T_z$, $T_x$, $T_y$, $R_z$, $R_x$, $R_y$.

**MetaRoom Dataset.** To implement the 3D-2D projection from point clouds to images through the camera, we first introduce the *MetaRoom* dataset based on Webots [34]. The *MetaRoom* dataset contains 20 commonly-seen indoor objects and for each object, we place it on a small table located in the center of an empty room with the size of $3\,\mathrm{m} \times 3\,\mathrm{m}$. For each object, the collected data is associated with the point cloud of the entire room, all the camera poses, and camera intrinsic parameters. We

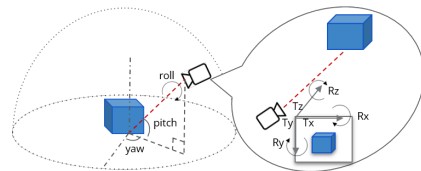

Figure 3: The illustration of coordinates in *MetaRoom* and real robot with camera motion.

first reconstruct the $0.0025\,\mathrm{m}$-density point clouds of the room together with each object on the table through random snapshots from an RGBD camera with 500 camera poses for training and 120 camera poses for testing. The objects are $0.1\,\mathrm{m}$ higher than the origin of the global coordinate. For the training set, the camera is oriented toward the origin and uniformly randomly moves within a semi-spherical area of $2.3\,\mathrm{m} \sim 2.9\,\mathrm{m}$ radius with the yaw range of $360°$, the pitch range of $0° \sim 90°$ and roll range of $-60° \sim 60°$. The poses in the test set are $2.8\,\mathrm{m}$ away from the original with 6 fixed yaw angles uniformly distributed in $0° \sim 360°$, the pitch angle of $10°$, and the roll angle of $0°$. Note that the poses in the training set are at least $15°$ away from any yaw, pitch, or roll angle of every pose in the test set to force the model to generalize rather than just memorizing the training data. The coordinate is illustrated in Fig. 3.

**Model Training.** We adopt two off-the-shelf representative architectures, ResNet-18 and ResNet-50 [4], to train base classifiers on *MetaRoom* for 100 epochs. For baselines, we adopt motion-specific augmentation as a defense to train the **Motion Augmented** models while **Vanilla** models are not trained with the augmentation since the augmented classifier could generalize well to the images with corresponding perturbation following [31, 50]. The motion augmentations are consistent with smoothing Gaussian distributions with $\sigma$ shown in Tab. 2. Here motion augmented and vanilla are different strategies to train a **Base** classifier for **Smoothed** model via motion smoothing in Def. 3.

**Evaluation Metrics.** The **Benign Accuracy** is calculated as the ratio of correctly classified projected images without any perturbation, which is used to show the robustness/accuracy trade-off [28]. Based on the literature on spatial robustness [53, 54] that gradient-based attack methods [17, 41] do not perform better than grid-search-based attacks for spatial adversarial samples due to the highly non-convex optimization landscape in semantic transformation space, we use grid search to evaluate the worst-case perturbations. We uniformly sample 5 and 100 perturbed camera motions within the radius and consider the model is not robust if any of them is wrongly classified, and report the ratio of robust ones over the whole test set as **5-perturbed Empirical Robust Accuracy** and **100-perturbed Empirical Robust Accuracy**, respectively. To provide a rigorous lower bound on the accuracy against any possible perturbations, we report **Certified Accuracy** [31, 50] to evaluate the certification results, which is the fraction of test images that are both correctly classified and satisfy the certification condition of (7) within motion perturbation radius through smoothing, meaning models will predict correctly with at least this certified accuracy for any camera perturbation within the given motion radius with high confidence. Following convention [28], we use the confidence of 99% and 1000 smoothing camera motions under zero-mean Gaussian distribution for test images.

## 4.2 Empirical Robustness against Camera Motion with Smoothing

We answer the question about the influence of camera motion perturbation on the perception performance empirically. For models trained with motion augmented defense, we compare the base

| Camera Motion Types | Motion Aug. ResNet18 | Motion Aug. ResNet50 |
|---|---|---|
| $T_z$, radius [-0.1m, 0.1m] | Base / Smoothed | Base / Smoothed |
| Benign Accuracy | **0.858** / 0.842 | **0.875** / 0.867 |
| 100-perturbed Emp. Robust Acc. | 0.758 / **0.817** | 0.775 / **0.825** |
| $T_x$, radius [-0.05m, 0.05m] | Base / Smoothed | Base / Smoothed |
| Benign Accuracy | **0.925** / 0.900 | **0.883** / 0.867 |
| 100-perturbed Emp. Robust Acc. | 0.785 / **0.867** | 0.633 / **0.800** |
| $T_y$, radius [-0.05m, 0.05m] | Base / Smoothed | Base / Smoothed |
| Benign Accuracy | **0.925** / 0.892 | 0.917 / **0.942** |
| 100-perturbed Emp. Robust Acc. | 0.808 / **0.842** | 0.842 / **0.908** |
| $R_z$, radius [-7°, 7°] | Base / Smoothed | Base / Smoothed |
| Benign Accuracy | 0.933 / **0.958** | 0.933 / **0.950** |
| 100-perturbed Emp. Robust Acc. | 0.875 / **0.892** | 0.867 / **0.917** |
| $R_x$, radius [-2.5°, 2.5°] | Base / Smoothed | Base / Smoothed |
| Benign Accuracy | **0.975** / 0.950 | 0.925 / **0.942** |
| 100-perturbed Emp. Robust Acc. | **0.908** / 0.892 | 0.850 / **0.917** |
| $R_y$, radius [-2.5°, 2.5°] | Base / Smoothed | Base / Smoothed |
| Benign Accuracy | 0.917 / **0.925** | 0.975 / **0.992** |
| 100-perturbed Emp. Robust Acc. | 0.867 / **0.925** | 0.933 / **0.983** |

Table 1: The comparison between base and motion smoothed models which are both trained with motion augmentations in terms of benign and 100-perturbed empirical robust accuracy for all camera motions. The higher one between each base and motion smoothed model is in **bold**.

and smoothed classifiers over benign and empirical robust accuracy in Tab. 1. It can be seen that under each motion perturbation, the empirical robust accuracy is lower than the benign accuracy for the base and smoothed models, showing that the perception models are not robust under motion perturbations even with the motion augmented defense when training. However, the gap between empirical robust accuracy and benign accuracy of motion smoothed models become less than those of base models due to the effective motion smoothing that improves robustness. Specifically, the motion smoothed models perform better in terms of empirical robust accuracy than the base models for camera motion perturbation along all axis. Interestingly, under a similar magnitude of radius, rotational smoothing will mostly increase the benign accuracy while translational smoothing tends to decrease the benign accuracy due to the robustness/accuracy trade-off [28]. The comparison and analysis between 5-perturbed and 100-perturbed empirical robust accuracy can be found in supplementary materials Section B.3 and Table 4.

### 4.3 Comparison of Certified Accuracy

Here we aim to show how we certify the robust accuracy using the motion smoothing strategy against motion perturbation, so we compare the smoothed augmented models with the smoothed vanilla models in Tab. 2. For the motion perturbations on each axis, under the same smoothing strategy, most smoothed augmented models have better performance in benign, empirical robust and certified accuracy compared to smoothed vanilla models, showing that the motion augmentation improves the overall perception performance, empirical and certifiable robustness. Besides, the certified accuracy is only slightly lower than empirical robust accuracy under the same perturbation radius, which shows that our certification guarantees are effective and scalable against all camera motion perturbations for different baseline models.

### 4.4 Ablation Study

**Certified accuracy under different radius.** Fig. 4 shows the certified accuracy with respect to the motion perturbation radius on each axis. We find that as the certified radius increases, the certified accuracy decreases and comes to 0 at some radius. Smoothed vanilla models have slowly decaying certified accuracy. For motion augmented models, the certified accuracies beyond the certified radii of *Translation z* and *Rotation y* still remain high until two times larger than $\sigma$ in motion smoothing, while *Translation y* and *Rotation x* have quickly decayed certified accuracies, showing that *Translation z* and *Rotation y* can be better certified over larger perturbation radii.

**Comparison of different model complexity.** Smoothed vanilla ResNet-18 performs better in certified accuracy compared to ResNet-50 within smoothing radii $\sigma$. With motion augmented defense, although most empirical robust accuracy of base ResNet-50 is worse than base ResNet-18 in Tab. 1, ResNet-50 has better certified accuracy after motion smoothing under larger perturbation radii for most 1-axis camera motions in Fig. 4, which indicates that motion-augmented smoothed classifiers

| $\sigma_z$ | $T_z$ within radius [-0.1m, 0.1m] | Benign Acc | 5-pert Emp. Acc | Certified Acc |
|---|---|---|---|---|
| | Smoothed Vanilla ResNet-18 | **0.858** | 0.817 | 0.792 |
| 0.1m | Smoothed Augmented ResNet-18 | 0.842 | **0.833** | **0.817** |
| | Smoothed Vanilla ResNet-50 | 0.675 | 0.617 | 0.558 |
| | Smoothed Augmented ResNet-50 | **0.867** | **0.850** | **0.817** |
| $\sigma_x$ | $T_x$ within radius [-0.05m, 0.05m] | Benign Acc | 5-pert Emp. Acc | Certified Acc |
| | Smoothed Vanilla ResNet-18 | 0.825 | 0.783 | 0.700 |
| 0.05m | Smoothed Augmented ResNet-18 | **0.900** | **0.875** | **0.833** |
| | Smoothed Vanilla ResNet-50 | 0.767 | 0.675 | 0.508 |
| | Smoothed Augmented ResNet-50 | **0.867** | **0.825** | **0.708** |
| $\sigma_y$ | $T_y$ within radius [-0.05m, 0.05m] | Benign Acc | 5-pert Emp. Acc | Certified Acc |
| | Smoothed Vanilla ResNet-18 | 0.850 | 0.825 | 0.767 |
| 0.05m | Smoothed Augmented ResNet-18 | **0.892** | **0.875** | **0.817** |
| | Smoothed Vanilla ResNet-50 | 0.792 | 0.767 | 0.683 |
| | Smoothed Augmented ResNet-50 | **0.942** | **0.925** | **0.892** |
| $\sigma_\theta$ | $R_z$ within radius [-7°, 7°] | Benign Acc | 5-pert Emp. Acc | Certified Acc |
| | Smoothed Vanilla ResNet-18 | 0.817 | 0.742 | 0.608 |
| 7° | Smoothed Augmented ResNet-18 | **0.958** | **0.933** | **0.883** |
| | Smoothed Vanilla ResNet-50 | 0.758 | 0.717 | 0.633 |
| | Smoothed Augmented ResNet-50 | **0.950** | **0.917** | **0.883** |
| $\sigma_\theta$ | $R_x$ within radius [-2.5°, 2.5°] | Benign Acc | 5-pert Emp. Acc | Certified Acc |
| | Smoothed Vanilla ResNet-18 | 0.842 | 0.800 | 0.633 |
| 2.5° | Smoothed Augmented ResNet-18 | **0.950** | **0.942** | **0.867** |
| | Smoothed Vanilla ResNet-50 | 0.767 | 0.742 | 0..567 |
| | Smoothed Augmented ResNet-50 | **0.942** | **0.933** | **0.867** |
| $\sigma_\theta$ | $R_y$ within radius [-2.5°, 2.5°] | Benign Acc | 5-pert Emp. Acc | Certified Acc |
| | Smoothed Vanilla ResNet-18 | 0.892 | 0.875 | 0.708 |
| 2.5° | Smoothed Augmented ResNet-18 | **0.925** | **0.925** | **0.917** |
| | Smoothed Vanilla ResNet-50 | 0.808 | 0.783 | 0.717 |
| | Smoothed Augmented ResNet-50 | **0.992** | **0.992** | **0.967** |

Table 2: The comparison between smoothed vanilla and smoothed augmented models in terms of benign, 5-perturbed empirical robust (5-perb Emp.), and certified accuracy for with all camera motions. The higher one between vanilla and motion augmented models is in **bold**.

| Model | Benign Accuracy | 10-perturbed Empirical Robust Accuracy | | | | | |
|---|---|---|---|---|---|---|---|
| | | $T_x$ | $T_y$ | $T_z$ | $R_x$ | $R_y$ | $R_z$ |
| Base | 83.3% | 76.3% | 78.1% | 79.8% | 77.1% | 74.6% | 82.5% |
| Smoothed | | **78.9%** | **81.6%** | **82.5%** | **77.2%** | **75.4%** | **83.3%** |

Table 3: Quantitative results of real-world robotic perception model. The perturbation range of translations $(T_x, T_y, T_z)$ is $[-1.25cm, 1.25cm]$ and perturbation range of rotations $(R_x, R_y, R_z)$ is $[-2.5°, 2.5°]$. The variance of zero-mean Gaussian smoothing distribution is $0.625cm$ translations and $1.25°$ for rotations. The higher values between the base and smoothed ones are in **bold**.

with larger complexity tend to be more certifiably robust to camera motion perturbations, although they may suffer from lower empirical robust accuracy due to overfitting.

## 4.5 Real-world Experiments

**Experiment setup.** We conduct hardware robotic experiments using the Kinova-Gen3 Arm of 7-DoF with an eye-in-hand camera in the pick-place task environment. These 6 objects with different shapes and colors and the perception model deployed on the arm is based on the ResNet50 classification model. For the data collection, following the spherical coordinate in Figure 3, 2500 images along the random trajectories are captured as training set and test set includes 114 test poses with 19 non-overlapped fixed poses for all 6 objects. Following the metrics in Section 4.1, for both the base model and smoothed model with variance $0.625cm$ for all translations and $1.25°$ for all rotations, we adopt 10 uniform samples over $[-1.25cm, 1.25cm]$ and $[-2.5°, 2.5°]$ as empirical robust accuracy. We report benign accuracy without any perturbation using the base model for comparison. More details can be found in supplementary materials Section B.4.

**Results and analysis.** From Table 3, it can be seen that the 10-perturbed empirical robust accuracy is lower than the benign accuracy for the base model, which means the small perturbations along

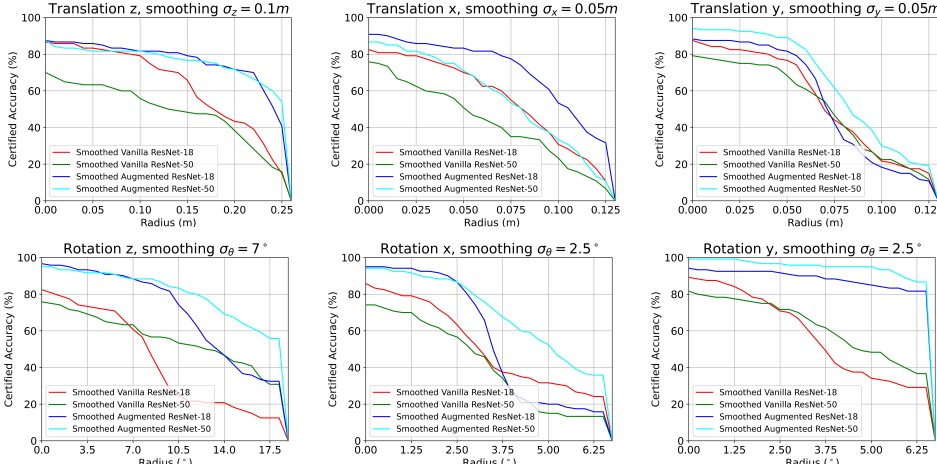

Figure 4: Certified accuracy with respect to radius for all ego-motions

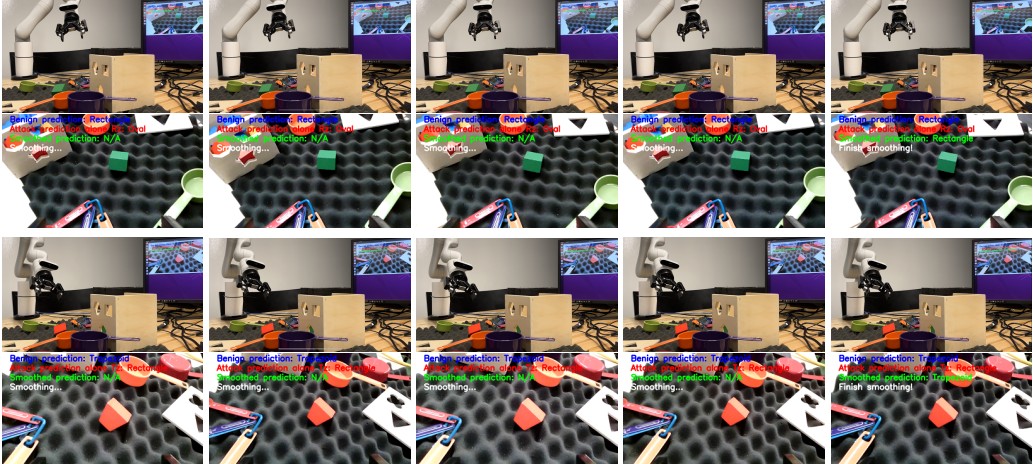

Figure 5: Smoothing process to improve robustness against camera motion of $R_z$ (top) and $T_z$ (bottom). The left four columns are randomized smoothing samples, and the right column is the classification result after smoothing.

each axis do influence the performance of the real-world robotic perception. Besides, our proposed smoothed model improves the robust accuracy against all the perturbations, and results of $T_z$ and $R_z$ after smoothing are very close to benign accuracy. Since the perturbations are small, the perturbation influence is not that significant. But the smoothing variance is also tiny enough to be reasonable and practical in robotic applications, the improvement is observable and eligible to validate the effectiveness of the smoothing method. See Figure 5 for qualitative results.

## 5 Conclusion

In this work, we study the robust visual perception against camera motion perturbation as the projective transformation from 3D to 2D. We propose a robustness certification framework via a camera motion smoothing approach to provide robust guarantees for image classification models for real-world robotic applications. We collect a realistic indoor robotic dataset MetaRoom with the dense point cloud map for robustness certification against camera motion perturbation. We conduct extensive experiments to compare the empirical robust accuracy with the certified robust accuracy for the motion smoothed model within large radii of 6-axis camera motion perturbations, i.e., translations and rotations, to guarantee the lower bounds of accuracy for trustworthy robotic perception. We further conduct real-world robot experiments to show the effectiveness of the smoothing method.

**Acknowledgments**

We gratefully acknowledge support from the National Science Foundation under grant CAREER CNS-2047454. We also would like to thank Prof. Bo Li from UIUC for the thoughtful feedback and Shiqi Liu from CMU for helping to conduct the real robot experiment.

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
