# OpenReview forum: "Robustness Certification of Visual Perception Models via Camera Motion Smoothing"
_robot-learning.org/CoRL/2022/Conference — CoRL 2022 Poster_

### Official Review · Reviewer_BZ5K · 2022-07-27

**Originality:** Good
**Technical Quality:** Good
**Clarity Of Presentation:** Very Good
**Impact:** 3

**Recommendation:**

Strong Reject: I recommend rejecting the paper and will argue for my recommendation even if other reviewers hold a different opinion.

**Summary:**

This paper addresses the sensitivity of visual perception model to camera perturbations.
Overall, although the paper seems to provide meaningful results to computer vision community, it is not well suited for CoRL audience. It essentially hits all the below desk rejection criteria:

* No robotics: This is pure a computer vision problem, and the connection to robotics is through the argument that perception is one of the tools that are used in robots. However, there is no robotics experiment involved and I found the connection to robotics as a stretch.

* An algorithm that was only evaluated in simulation without credible evidence on the possibility of transfer to a real robot learning due to sim2real problems or data efficiency. The images from MetaRoom looks far from being close a realistic real-world robotics tasks.

**Issues:**

Out of scope for CoRL due to the lack of any robotics experiment or evidence/results on how this could improve a robot learning problem.

**Quality Of The Limitations Section:**

Limitations are not well addressed

**Reviewer Expertise:**

4: The reviewer is confident but not absolutely certain that the evaluation is correct

**Robotics Focus:**

Irrelevant to robotics

**Strengths And Weaknesses:**

Please refer to the summary.

**Summary Of Recommendation:**

Out of scope for CoRL.

---

> ### Author Response · Authors · 2022-08-22
> **Thanks for your valueable comments and we have conducted real-world robot experiments to validate our method**
>
> **Comment:**
>
> We thank the reviewer for recognizing our work novel and really appreciate the reviewer's suggestions on hardware experiments to help improve the quality of our work. We have conducted the real-world robot experiment, answered the questions below and improved our paper following the suggestions. Feel free to check out the updated main text with the appendix and all the major changes are marked in blue. The new video demo for real-world robot experiments can be found in the zip file or on anonymous link [https://drive.google.com/file/d/1Zy3oQUHk-EVulA3ma7CKnrhTPrydDVli/view?usp=sharing](https://drive.google.com/file/d/1Zy3oQUHk-EVulA3ma7CKnrhTPrydDVli/view?usp=sharing). Looking forward to your response and further discussion.
>
> > **Q1:** No robotics: This is pure a computer vision problem, and the connection to robotics is through the argument that perception is one of the tools that are used in robots. However, there is no robotics experiment involved and I found the connection to robotics as a stretch.
>
> Thanks for your comments and suggestion. We have conducted a hardware robotics experiment with a robot arm and eye-in-hand camera for image classification, validating the effectiveness of the smoothed model against camera motion perturbation from the end of a robotic arm in real-world robotic applications, e.g. the perturbation of camera for robot manufacturing. We have updated the environment setup and result analysis in Abstract, Section 1, Section 4.5 and Appendix B.4, Figure 2, Figure 5, Figure 7 and Table 3. The video demo for the real-world robot experiment can be found in the zip file or on anonymous link [https://drive.google.com/file/d/1Zy3oQUHk-EVulA3ma7CKnrhTPrydDVli/view?usp=sharing](https://drive.google.com/file/d/1Zy3oQUHk-EVulA3ma7CKnrhTPrydDVli/view?usp=sharing). Hope it will help to dismiss your concern.
>
> > **Q2:** An algorithm that was only evaluated in simulation without credible evidence on the possibility of transfer to a real robot learning due to sim2real problems or data efficiency. The images from MetaRoom looks far from being close a realistic real-world robotics tasks.
>
> Thanks for your comment. We build up the indoor environment based on realistic simulation Webots for better point cloud reconstruction. To compensate for the weakness of non-robotics tasks from this simulation environment, we conduct the real-world robot experiment to test empirical robustness against the motion of the eye-in-hand camera with a robot arm, which is commonly seen in real-world robotics tasks. Therefore the proposed method in the paper can be transferred to real robot problems for robustness and trustworthiness.
>
> > **Q3:** Out of scope for CoRL due to the lack of any robotics experiment or evidence/results on how this could improve a robot learning problem.
>
> Thanks for your comment. We have added real robotics experiments to show the effectiveness of our method in robotic applications. Our results also show that the smoothed model is more robust than the base perception model against the commonly-seen camera perturbation, which helps to improve the robust robot learning problem with adversarial machine learning.
>
> **Zip File:**
>
> /attachment/0169e007cda66a1cb0b32275b5da66278883e14f.zip

---

> > ### Comment · Reviewer_BZ5K · 2022-08-23
> > **Comment**
> >
> > Thanks for conducting the robot experiment. I totally see the value of such experiment for the vision community, but the shown robotic experiment is merely about using robots to take new photos for classification, rather than showing the impact of image smoothness on improving the robot performance during an execution of a robotic tasks such as grasping, placing etc. It would be great to compare these two scenarios and show:
> > Scenario A: Train a behavior cloning or reinforcement learning policy with the raw image data and show the policy achieved N% success.
> > Scenario B: Apply the proposed image smoothing technique and show that with the same amount of data the robot can achieve M% success, while M > N.

---

> > > ### Author Response · Authors · 2022-08-24
> > > **Response to further comment**
> > >
> > > Thanks for the prompt response and additional comments. Thank you for recognizing the value of our work in real-world robotic perception. We humbly argue that the main contribution of this paper is the provable theoretical analysis of camera motion smoothing method to certify the robustness of deep learning-based image classification model, demonstrating its effectiveness to improve the robustness against the camera perturbation both in simulation and real-world robotic perception. Robotic Perception is an indispensable module for the non-end-to-end robotic pipeline including planning, control, etc, so the robustness against unexpected perturbation of the camera is the preliminary of safe and trustworthy robot execution and decision-making.
> > >
> > > Regarding the suggestions on comparing the two scenarios using behavior cloning or reinforcement learning for end-to-end robotic tasks, we think highly of these valuable ideas as an extension of our certification framework to certify the robustness of such end-to-end robot executions. However, the formulation of behavior cloning or reinforcement learning is essentially different from the formulation of robotic perception. Besides, provable robustness certification for reinforcement learning has been barely studied in robust machine learning until recent works [1, 2], which still focus on simple tabular cases with l-p bounded attacks. Therefore, conducting image smoothing for a behavior cloning or reinforcement learning policy for robot execution tasks needs more in-depth theoretical analysis and is out of scope of the focus of this work. Nevertheless, we agree that it is definitely an interesting and impactful topic, and we regard it as a future work to certify the robustness of robotic decision-making models based on our method.
> > >
> > > Looking forward to your feedback and further discussion, thanks again for your insightful and valuable suggestion!
> > >
> > >
> > > [1] Wu, F., Li, L., Huang, Z., Vorobeychik, Y., Zhao, D., & Li, B. (2021, September). CROP: Certifying Robust Policies for Reinforcement Learning through Functional Smoothing. In International Conference on Learning Representations. 2022
> > >
> > > [2] Wu, J., & Vorobeychik, Y. (2022, June). Robust Deep Reinforcement Learning through Bootstrapped Opportunistic Curriculum. In International Conference on Machine Learning (pp. 24177-24211). PMLR.

---

> ### Author Response · Authors · 2022-08-26
> **Post rebuttal discussion**
>
> We sincerely thank the reviewer for your previous insightful questions and suggestions, and we have tried our best to add additional experiments/clarifications to our paper as well as answer the questions. Please let us know if you have further questions or comments. We really look forward to your feedback to further improve our work. Thank you!

---

### Official Review · Reviewer_C4we · 2022-07-31

**Originality:** Fair
**Technical Quality:** Good
**Clarity Of Presentation:** Fair
**Impact:** 2

**Recommendation:**

Weak Reject: I recommend rejecting the paper, but will not argue for my recommendation if the majority of other reviewers have a different opinion.

**Summary:**

This paper addresses an important problem - robustness of perception models to input noise introduced by camera motion. The authors consider consider the problem of image classification on images generated from point clouds, and the camera motion noise is considered in the point cloud sensor. The bounded noise in the point cloud input can then be mapped to bounds on the projected image. Then training a smoothed classifier for the image enables provable robustness bounds on the input camera noise.

**Issues:**

See my comments in strengths and weaknesses

**Quality Of The Limitations Section:**

Limitations are addressed clearly

**Reviewer Expertise:**

4: The reviewer is confident but not absolutely certain that the evaluation is correct

**Robotics Focus:**

Sufficient demonstration on hardware

**Strengths And Weaknesses:**

Strengths
- Robustness of perception models to hardware noise is a very important problem to consider
- Approaching it from a certification perspective makes sense considering the importance of safety guarantees for robotics applications


Weaknesses
- The use of "camera" and "image" is very confusing  when the certification is applied to point cloud inputs (which are projected into an image, but the noise is still captured in the point clouds).
- Lemma 1 is not used anywhere in the paper from what I can tell. Perhaps it can be moved to the supplementary materials
- More background can be provided for the smoothed classifiers. It's a key component of the pipeline and can benefit from more introduction. In particular, equation 8 uses a key result of smoothed classifiers to obtain certification guarantees without providing the background.
- Line 190-192 "The robustness can be evaluated as the classification accuracy under worst-case perturbations. We uniformly sample 5 perturbed camera motions within the radius and consider the model is not robust." This metric seems a bit meaningless to me, if we take any classifier and sample random minor perturbations, the output is unlikely to change. I believe a better empirical evaluation is to perform an adversarial attack against the model and that should demonstrate a significant gap between the smoothed model and the vanilla model.
- In my opinion, the work is incremental. I see it as an application of training smoothed classifiers for images and propagating the guarantees to the camera extrinsics.

**Summary Of Recommendation:**

This paper definitely studies an important problem in certifying the robustness of perception models due to sensor motion and noise in camera extrinsics. However I believe the manuscript and experimentation has a lot of room for improvement:
- More clear delivery, I found the problem formulation and terminology quite confusing and the diagrams were not the most informative
- More background can be provided on smoothed classifiers, since it's an integral part of the method
- The empirical robust accuracy is evaluated by sampling 5 random perturbations and does not seem like a meaningful metric

---

> ### Author Response · Authors · 2022-08-22
> **Thanks for your valuable comments and we have add more additional experiments and clarification to dismiss your concern (1/2)**
>
> **Comment:**
>
> We thank the reviewer for recognizing our work as novel and important and really appreciate the reviewer's suggestions to help improve the quality of our work. We answered the questions below and improved our paper following the suggestions. Feel free to check out the updated main text with the appendix and all the major changes are marked in blue. The new video demo for real-world robot experiments can be found in the zip file or on  [anonymous link](https://drive.google.com/file/d/1Zy3oQUHk-EVulA3ma7CKnrhTPrydDVli/view?usp=sharing). Looking forward to your response and further discussion.
>
> > **Q1:** The use of "camera" and "image" is very confusing when the certification is applied to point cloud inputs (which are projected into an image, but the noise is still captured in the point clouds).
>
> Sorry for the confusion and thanks for your question. In our certification framework for image classification, the input is always images, which are projected from a fixed point cloud through the camera's movement. Therefore, the noise comes from the camera motion while the point clouds are given as an oracle and do not change. Recent work [1] shows the certification of point clouds as input. Thanks again for the question and we have clarified it in the fourth paragraph of Section 1 in the updated paper.
>
> > **Q2:** Lemma 1 is not used anywhere in the paper from what I can tell. Perhaps it can be moved to the supplementary materials
>
> Thank you for pointing it out. We further clarify that Lemma 1 is derived from Definition 2 and serves as the foundation for Theorem 1. Therefore, we have revised Section 3.3 to make it coherent and consistent in the updated paper.
>
> > **Q3:** Line 190-192 "The robustness can be evaluated as the classification accuracy under worst-case perturbations. We uniformly sample 5 perturbed camera motions within the radius and consider the model is not robust." This metric seems a bit meaningless to me, if we take any classifier and sample random minor perturbations, the output is unlikely to change. I believe a better empirical evaluation is to perform an adversarial attack against the model and that should demonstrate a significant gap between the smoothed model and the vanilla model.
>
> Thanks for your suggestion. Based on the literature review on robustness and adversarial learning for **spatial or semantic transformation** [4,5], it is shown that gradient-based attack methods like FGSM or PGD perform worse than grid-search-based attacks to find adversarial samples due to highly non-convex optimization landscape in semantic transformation space, i,e, there are countless local minimums for the problem that can hardly solved by gradient-based methods.
>
> As a result, we adopt the grid search to find the worst-case perturbations instead of using gradient-based attacks. In order to compare the effect of the grid search, we conduct further experiments by uniformly sampling 100 samples besides 5 samples and report the ratio of robust ones over the whole test set, which is stronger than 5-perturbed empirical robustness accuracy. We have updated Section 4.1, Appendix B.3 and Table 1, 7, 8. Here is the comparison of 100-perturbed and 5-perturbed robustness for smoothed classifiers. Hope it can help to address your concern.
>
> |       Camera Motion Types      |   Smoothed ResNet18   |    Smoothed ResNet50  |
> |:------------------------------:|:---------------------:|:---------------------:|
> |    Tz, radius [-0.1m, 0.1m]    | Vanilla / Motion Aug. | Vanilla / Motion Aug. |
> |  5-perturbed Emp. Robust Acc.  |     0.817 / 0.833     |     0.617 / 0.850     |
> | 100-perturbed Emp. Robust Acc. |     0.783 / 0.817     |     0.567 / 0.825     |
> |   Tx, radius [-0.05m, 0.05m]   | Vanilla / Motion Aug. | Vanilla / Motion Aug. |
> |  5-perturbed Emp. Robust Acc.  |     0.783 / 0.875     |     0.675 / 0.825     |
> | 100-perturbed Emp. Robust Acc. |     0.758 / 0.867     |     0.617 / 0.800     |
> |   Ty, radius [-0.05m, 0.05m]   | Vanilla / Motion Aug. | Vanilla / Motion Aug. |
> |  5-perturbed Emp. Robust Acc.  |     0.825 / 0.875     |     0.767 / 0.925     |
> | 100-perturbed Emp. Robust Acc. |     0.792 / 0.842     |     0.758 / 0.908     |
> |      Rz, radius [-7°, 7°]      | Vanilla / Motion Aug. | Vanilla / Motion Aug. |
> |  5-perturbed Emp. Robust Acc.  |     0.742 / 0.933     |     0.717 / 0.917     |
> | 100-perturbed Emp. Robust Acc. |     0.717 / 0.892     |     0.675 / 0.917     |
> |    Rx, radius [-2.5°, 2.5°]    | Vanilla / Motion Aug. | Vanilla / Motion Aug. |
> |  5-perturbed Emp. Robust Acc.  |     0.800 / 0.942     |     0.742 / 0.933     |
> | 100-perturbed Emp. Robust Acc. |     0.750 / 0.892     |     0.692 / 0.917     |
> |    Ry, radius [-2.5°, 2.5°]    | Vanilla / Motion Aug. | Vanilla / Motion Aug. |
> |  5-perturbed Emp. Robust Acc.  |     0.875 / 0.925     |     0.783 / 0.992     |
> | 100-perturbed Emp. Robust Acc. |     0.808 / 0.925     |     0.742 / 0.983     |
>
>
>
> **Zip File:**
>
> /attachment/1944c9f38ee2988c5e975d6d0db067a3b63d49ac.zip

---

> > ### Author Response · Authors · 2022-08-22
> > **Thanks for your valuable comments and we have add more additional experiments and clarification to dismiss your concern (2/2)**
> >
> > **Comment:**
> >
> > > **Q4:** More background can be provided for the smoothed classifiers. It's a key component of the pipeline and can benefit from more introduction. In particular, equation 8 uses a key result of smoothed classifiers to obtain certification guarantees without providing the background.
> >
> > Thanks for your suggestion. We fully agree that the smoothed classifier is the key component to achieving the certification goal in Eq. (4). Prior works [1,2,3] show that a smoothed model can be certified using equation (8) if the image transformation is resolvable. To this end, we first give Lemma 1 to show the relative projection is generally compatible with the global projection, which indicates that image projection can be regarded as a resolvable transformation and applied the resolvable certification condition from [1,2]. We have revised Section 3.3 to add more background for the certification in the updated version.
> >
> > > **Q5:** In my opinion, the work is incremental. I see it as an application of training smoothed classifiers for images and propagating the guarantees to the camera extrinsics.
> >
> > Thanks for your comment. Although our work is under the umbrella of certifiable robustness and randomized smoothing, we humbly argue its contribution and novelty in bridging the semantic transformation (image projection) robustness theory with the motion of the camera sensor for robust robotic perception. We believe this is non-trivial and is an important step to apply the robustness certification theory into practical robotic applications.
> >  Furthermore, we conducted the physical hardware real-world robot experiment with a robotic arm to validate the smoothed model in robotic applications. We have updated Section 4.5, Appendix B.4 and the robot experiment demo is available in the zip file or on an anonymous link [https://drive.google.com/file/d/1Zy3oQUHk-EVulA3ma7CKnrhTPrydDVli/view?usp=sharing](https://drive.google.com/file/d/1Zy3oQUHk-EVulA3ma7CKnrhTPrydDVli/view?usp=sharing).
> >
> > > **Q6:** More clear delivery, I found the problem formulation and terminology quite confusing and the diagrams were not the most informative
> >
> > Sorry for the confusion and thanks for your suggestion. We have updated Section 1 and 3.2 for further clarification on problem formulation and method terminology.
> >
> > > **Q7:** More background can be provided on smoothed classifiers, since it's an integral part of the method
> >
> > Thanks for your comment. We have updated Section 3.3 to provide background on robustness certification for semantic transformation and make it coherent.
> >
> > > **Q8:** The empirical robust accuracy is evaluated by sampling 5 random perturbations and does not seem like a meaningful metric
> >
> > Thanks for your comment. We have done the further experiment with the metric of sampling 100 random perturbations and updated Section 4.1, Appendix B.3 and Table 1, 7, 8.
> >
> > --
> >
> > [1] Chu, W., Li, L., & Li, B. (2022). TPC: Transformation-Specific Smoothing for Point Cloud Models. arXiv preprint arXiv:2201.12733.
> >
> > [2] Li, L., Weber, M., Xu, X., Rimanic, L., Kailkhura, B., Xie, T., ... & Li, B. (2021, November). Tss: Transformation-specific smoothing for robustness certification. In Proceedings of the 2021 ACM SIGSAC Conference on Computer and Communications Security (pp. 535-557).
> >
> > [3] Hao, Z., Ying, C., Dong, Y., Su, H., Song, J., & Zhu, J. (2022, June). GSmooth: Certified Robustness against Semantic Transformations via Generalized Randomized Smoothing. In International Conference on Machine Learning (pp. 8465-8483). PMLR.
> >
> > [4] Engstrom, L., Tran, B., Tsipras, D., Schmidt, L., & Madry, A. (2019, May). Exploring the landscape of spatial robustness. In International conference on machine learning (pp. 1802-1811). PMLR.
> >
> > [5] Sitawarin, C., Golan-Strieb, Z. J., & Wagner, D. (2022, June). Demystifying the Adversarial Robustness of Random Transformation Defenses. In International Conference on Machine Learning (pp. 20232-20252). PMLR.
> >
> > **Zip File:**
> >
> > /attachment/e164d9e5a01f62b8d80c40e6b7c46b3fad4dcb3c.zip

---

> ### Author Response · Authors · 2022-08-26
> **Post rebuttal discussion**
>
> We sincerely thank the reviewer for your previous insightful questions and suggestions, and we have tried our best to add additional experiments/clarifications to our paper as well as answer the questions. Please let us know if you have further questions or comments. We really look forward to your feedback to further improve our work. Thank you!

---

### Official Review · Reviewer_MJ8E · 2022-08-01

**Originality:** Good
**Technical Quality:** Excellent
**Clarity Of Presentation:** Excellent
**Impact:** 4

**Recommendation:**

Strong Accept: I recommend accepting the paper and will argue for my recommendation even if other reviewers hold a different opinion.

**Summary:**

The paper tackles the problem of ensuring robustness certification of a visual perception model, under camera perturbations. This is defines as a requirement that the model output shouldn't change, for camera perturbations, within a bounded region. The proposed solution is a simple smoothing technique. The paper provides for a theoretical result establishing robustness certification. The experimental results appear to be solid and validate the idea. The problem considered is very timely and pertinent, and is addressed with sufficient depth.

**Issues:**

Minor comments:
1. $\mathcal{Z}_{\alpha}$ being the set of all camera motion as $\alpha$ also denoting an arbitrary element in the set, is confusing notation.
2. In definition 2, $\subset \mathbb{R}^6$  seems like a typo.
3. Certification definition in (4), there seems to be a mistake. If one has to find Z_radius as well (which is a subset of Z_alpha) then a trivial solution would suffice, wouldn't it?

**Quality Of The Limitations Section:**

Limitations are addressed clearly

**Reviewer Expertise:**

5: The reviewer is absolutely certain that the evaluation is correct and very familiar with the relevant literature

**Robotics Focus:**

Highly relevant to robotics but no hardware experiments

**Strengths And Weaknesses:**

The paper is well written. Analyzed to good depth. Good theoretical content and solid experiments to justify the claims.

**Summary Of Recommendation:**

The paper is a solid contribution in the evolving field of certifiable perception. It makes good contribution. The paper analyzes the problem considered in good depth, provides solid theoretical results as well as experimental results, enough to justify the claim. This is a strong accept.

---

> ### Author Response · Authors · 2022-08-22
> **Thanks for your valueable comments and we have made revisions according to your suggestions**
>
> **Comment:**
>
> We thank the reviewer for recognizing our work as novel and solid and really appreciate the reviewer's suggestions to help improve the quality of our work. We answered the questions below and improved our paper following the suggestions. Feel free to check out the updated main text with the appendix and all the major changes are marked in blue. The new video demo for real-world robot experiment can be found in the zip file or on anonymous link [https://drive.google.com/file/d/1Zy3oQUHk-EVulA3ma7CKnrhTPrydDVli/view?usp=sharing](https://drive.google.com/file/d/1Zy3oQUHk-EVulA3ma7CKnrhTPrydDVli/view?usp=sharing).
>
> > **Q1:** Z_\alpha being the set of all camera motion as  \alpha also denoting an arbitrary element in the set, is confusing notation.
>
> Sorry for the confusion and thank you for pointing it out. We have revised the paper by  replacing all Z_alpha with Z to avoid the confusion.
>
> > **Q2:** In definition 2, R^6 seems like a typo.
>
> Thank you for pointing it out. We have fixed the typo by replacing R^6 with R^(C+3).
>
> > **Q3:** Certification definition in (4), there seems to be a mistake. If one has to find Z_radius as well (which is a subset of Z_alpha) then a trivial solution would suffice, wouldn't it?
>
> Thanks for your interesting question. If the goal is to find any Z_radius such that any element in Z_radius can suffice (4), then it would be a trivial solution. But for the certification in the literature, we need to find the Z_radius such that almost all elements in it can suffice (4) with a large confidence, where such Z_radius is non-trivial. That is to say, the certification goal is not to solve the *exist* problem of Z_radius but to solve the *forall* problem of Z_radius, i.e. given a Z_radius, verify all of its elements to see if they suffice (4) with high confidence (usually over 99%). We have clarified it in Section 3.2 in the updated version.
>
> **Zip File:**
>
> /attachment/ac6afa4709602e6ead001a99775431d32b3d431b.zip

---

> ### Author Response · Authors · 2022-08-26
> **Post rebuttal discussion**
>
> We sincerely thank the reviewer for your previous insightful questions and suggestions, and we have tried our best to add additional experiments/clarifications to our paper as well as answer the questions. Please let us know if you have further questions or comments. We really look forward to your feedback to further improve our work. Thank you!

---

### Official Review · Reviewer_vwtf · 2022-09-09

**Originality:** Very Good
**Technical Quality:** Very Good
**Clarity Of Presentation:** Fair
**Impact:** 4

**Recommendation:**

Weak Accept: I recommend accepting the paper, but will not argue for my recommendation if the majority of other reviewers have a different opinion.

**Summary:**

The paper presents a theory and an algorithm for certifying the robustness of a 2D image classifier against 3D camera motion disturbances. Building off of a series of studies on randomized smoothing [22, 25, 41, 43] that focus on perturbations in 2D image or 3D point cloud, the paper connects perturbation from 3D camera motion to 2D semantic perception through the projective relationship between colored 3D point cloud and its corresponding RGB image. The paper shows that by augmenting image classification training with the proposed random smoothing algorithm, the resulting classifier has provable robustness bound under camera motion perturbations.

**Issues:**

See my comments on weaknesses.

**Quality Of The Limitations Section:**

Additional details required

**Reviewer Expertise:**

4: The reviewer is confident but not absolutely certain that the evaluation is correct

**Robotics Focus:**

Sufficient demonstration on hardware

**Strengths And Weaknesses:**

Strengths:
- Certifiable perception is of great importance to the general robotics community. The paper extends the random smoothing theory to perturbations camera motion that is potentially more practical in common robotics settings, e.g., object navigation, mobile manipulation, and on-board sensing for drones.
- The problem setting of certifiable robustness of 2D perception under 3D camera rotation & translation is new.
- The algorithm is practical and conceptually simple.

Weaknesses:
- The method section is hard to follow. I think the main reason is the lack of intuitive explanation / illustration of basic concepts such as random smoothing and mapping noises from 3D motion to 2D image space. My suggestion is to provide an intuitive explanation of the basic components and intuition of the proposed algorithm, and the strategies that the section will take to prove the correctness of the algorithm. The current Figure 1 is somewhat confusing. I would suggest further polish Figure 1 and cross-reference it in the Method section to enhance the exposition.
- No quantitative evaluation for real robot experiment.
- Practically speaking, how would one generate the perturbed training set without having access to the data generation process? This seems to be a big assumption that should be addressed in the limitation section.
- I would appreciate more discussion on how to extend the framework to tasks beyond classification, e.g., segmentation, object detection, keypoint detection, depth estimation, which I believe would be of great interest to the robotics / geometric perception community.


**Summary Of Recommendation:**

Overall I believe the technical contribution is strong. But I would strongly urge the authors to rewrite part of the paper to further enhance the exposition. For details, see my strengths & weaknesses comments.

---

### Author Response · Authors · 2022-08-22
**General response**

**Comment:**

We thank all the reviewers and AC for their time and valuable suggestions. Following the suggestions from the reviewers, we have conducted additional experiments, especially real-world experiments, corrected some typos, and made the illustration clearer in our revision. We will list our major updates below.

- We added the real-world robot experiment in Abstract, Section 1, Section 4.5 and Appendix B.4, Figure 2, Figure 5, Figure 7 and Table 3.

- We added the 100-perturbed empirical robust accuracy as a stronger metric against camera motion perturbation and added the experiment results in Section 4.1, Appendix B.3, Table 1, Table 7, Table 8.

- We added the clarification of certification goal, background and terminology in Section 1, Section 3.2, 3,3.

All of our revisions are uploaded to OpenReview and highlighted in blue. A video demo for real-world robot experiments can be found in the zip file or on an anonymous link [https://drive.google.com/file/d/1Zy3oQUHk-EVulA3ma7CKnrhTPrydDVli/view?usp=sharing](https://drive.google.com/file/d/1Zy3oQUHk-EVulA3ma7CKnrhTPrydDVli/view?usp=sharing). We would be happy to discuss if there are any other concerns about our work. Thanks again for the suggestions and comments.

**Zip File:**

/attachment/dbcbfa8ecf6ac3cf208770fd4bbd751923865b39.zip

---

### Meta-Review · Area_Chair_UAZR · 2022-08-13

**Recommendation:** Accept (Poster)
**Confidence:** 2

**Metareview:**

This paper addresses the problem of ensuring robustness certification of a visual perception model, under camera perturbations via a smoothing technique.

The major contribution is the theoretical analysis to establish robustness certification for visual perception under camera perturbations.

The authors addressed the reviewers' concerns in a thorough manner, including a real-world robotic experiment which demonstrates the value to the robotics learning community.



**Best Paper Nomination:**

No

---

> ### Author Response · Authors · 2022-08-22
> **Thanks for your valuable comments and we have responsed to all the comments from the reviewers to improve our work**
>
> **Comment:**
>
>  Thanks for your valuable suggestions and for recognizing our work as an important problem. We also thank all the reviewers for recognizing our work’s novelty and really appreciate the reviewer's suggestions to help improve the quality of this paper.
>
> We have updated the main paper and appendix following the suggestions by all the reviewers, where the major changes are marked in blue. Specifically, to show a better link to robotics, we conduct a real-world robot experiment with a robot arm to validate our method.   We have also attached a video demo on a real robotic arm to show the application of our method, which can also be found on anonymous link [https://drive.google.com/file/d/1Zy3oQUHk-EVulA3ma7CKnrhTPrydDVli/view?usp=sharing](https://drive.google.com/file/d/1Zy3oQUHk-EVulA3ma7CKnrhTPrydDVli/view?usp=sharing).
>
> **Zip File:**
>
> /attachment/430c72d7e2ef9b311d58bfe83271aa6d046a970d.zip